# Multi-Focus Microscopy Image Fusion Based on Swin Transformer Architecture

**Han Hank Xia [1,2], Hao Gao [3,*], Hang Shao [2], Kun Gao [2] and Wei Liu [2]**

1. School of Electronic Science and Engineering, Nanjing University, Nanjing 210093, China; xiahannju@gmail.com
2. Yangtze Delta Region Institute of Tsinghua University, Jiaxing 314006, China; shaohang@zfti.org.cn (H.S.); tongyoung9123@163.com (K.G.); liuwei@zfti.org.cn (W.L.)
3. Institute of Advanced Technology, Nanjing University of Posts and Telecommunications, Nanjing 210023, China
* Correspondence: tsgaohao@gmail.com

**Abstract:** In this study, we introduce the U-Swin fusion model, an effective and efficient transformer-based architecture designed for the fusion of multi-focus microscope images. We utilized the Swin-Transformer with shifted window and path merging as the encoder for extracted hierarchical context features. Additionally, a Swin-Transformer-based decoder with patch expansion was designed to perform the un-sampling operation, generating the fully focused image. To enhance the performance of the feature decoder, the skip connections were applied to concatenate the hierarchical features from the encoder with the decoder up-sample features, like U-net. To facilitate comprehensive model training, we created a substantial dataset of multi-focus images, primarily derived from texture datasets. Our modulators demonstrated superior capability for multi-focus image fusion to achieve comparable or even better fusion images than the existing state-of-the-art image fusion algorithms and demonstrated adequate generalization ability for multi-focus microscope image fusion. Remarkably, for multi-focus microscope image fusion, the pure transformer-based U-Swin fusion model incorporating channel mix fusion rules delivers optimal performance compared with most existing end-to-end fusion models.

**Keywords:** multi-focus fusion; microscope images; Swin Transformer



## 1. Introduction

All optical imaging systems, especially for light microscopy, have a limited depth of field. Three-dimensional objects under investigation are often thicker than the depth of field of the imaging system, meaning that it is impossible to acquire a whole object completely in focus in one single image; only those portions that lie within the depth of field appear in focus and sharp, whereas the remaining regions are blurred by the system's point spread function (PSF) [1]. To overcome this limitation, multi-focus image fusion (MFIF) is an effective way to generate an all-in-focus image from a set of partially focused images, extending the depth of field of cameras. Compared with other methods to extend the depth of field of cameras, multi-focus image fusion tries to ensure the accuracy of the information, which is quite important for medical images. However, it takes some time to collect a set of partially focused images.

MFIF has been applied to various applications, such as micro-image fusion [2], visual sensor networks [3], visual power patrol inspection [4] and optical microscopy [5]. Image fusion technology has been developed for more than 30 years, during which various methods have been published. Conventional MFIM methods can be divided into spatial-based methods and transform-domain-based methods. Spatial-based methods fuse the image in the spatial domain directly, which can be further divided into pixel-based [6], block-based [7] and region-based [8] methods. In contrast, transform-domain-based methods

transform images into another domain firstly. Then, the transformed coefficients are merged by a pre-designed fusion rule. Finally, the fused image is reconstructed by applying the corresponding inverse transform based on the fused coefficients. Many transform-domain-based methods have been proposed so far, such as parse representation (SR) methods [9–11], multi-scale methods [12–16], gradient domain-based methods [17,18] and hybrid methods [19].

In recent years, with the advancement of deep learning, machine learning algorithms have been widely employed for various image fusion tasks, achieving remarkable success in the image fusion field. Various deep learning models, such as CNNs [20,21], GANs [22] and ensemble learning [23], have demonstrated their capability to attain state-of-the-art (SOTA) results in MFIF tasks. Recently, the transformer, a prominent architecture in natural language processing (NLP) [24], has been introduced into the computer vison domain to address the limitation of the long-range dependencies for CNN-based models [25–30]. Furthermore, models based on the Swin Transformer [31] achieved SOTA performance in various tasks, such as image classification, object detection and semantic segmentation. Swin-Unet, a novel pure transformer-based U-shaped encoder–decoder architecture, was proven to have excellent performance and generalization ability for medical image segmentation [32].

In summary, the transformer architecture based on the self-attention mechanism has achieved great success in the natural language processing domain. Recently, many transformer architectures have been introduced into the computer vision domain and achieved SOTA performance for various of computer vision tasks. In these studies, the Swin Transformer is an excellent work, which achieved perfect performance with linear computational complexity concerning image size. Therefore, we propose end-to-end image fusion models based on the Swin Transformer backbone to leverage the capabilities of a transformer for multi-focus image fusion in microscope images in this study. The main contributions of this paper can be summarized as follows:

1. Propose end-to-end models that use the Swin Transformer as a backbone to directly generate the fully focused images from multi-focus microscope images. Unlike existing CNN-based methods, this approach can extract long-range dependencies to generate naturally fully focused images.
2. To facilitate comprehensive model training, we created a substantial dataset of multi-focus images, primarily derived from texture datasets.
3. Propose two types of fusion rules which can be used in multi-focus image fusion, known as simple mix and channel mix. The evaluation results demonstrate that our transformer-based U-shaped models achieve state-of-the-art performance in multi-focus image fusion (MFIF) tasks for microscope images.

## 2. Related Work

### 2.1. Traditional MFIF Method

Traditional MFIF methods are typically categorized into two main groups: transform domain methods and spatial domain methods [33]. Transform domain methods mainly operate the decomposition coefficient after image transformation, encompassing three key fusion stages: image transformation, coefficient after image transformation, and inverse transformation reconstruction [34]. According to the application of the image transform, transform domain methods can be further classified into multi-scale decomposition (MSD)-based methods (e.g., Laplacian pyramid [14,35], discrete wavelet transform [36,37], nonsubsampled contourlet transform [38–40], neighbor distance filtering [41,42], empirical mode decomposition [43]), sparse representation (SR)-based methods (orthogonal matching pursuit [44,45]), gradient domain (GD)-based methods (structure tensor [46,47]), methods based on other transform (independent component analysis [48], cartoon-texture decomposition [49]) and methods based on the combination of different transforms (curvelet transform and wallet transform [50]). In the spatial domain methods, as the name suggests, source images are fused in the spatial domain based on the spatial features of images,

which can be divided into block-based methods [51–53], region-based methods [8,54] and pixel-based methods [55,56].

### 2.2. Convolutional Neural Network

In contrast to the traditional MFIF methods, deep-learning-based MFIF methods, especially the CNN-based MFIF method, can extract deep feathers to generate robust fully focused images with various input. Deep-learning-based MFIF methods can be categorized into decision-based and end-to-end methods [57]. In decision-based methods, firstly, a decision map that indicates the focus level (or activity level) is generated based on deep feathers from the CNN. Subsequently, post-processing steps are performed to generate the fused image according to the decision map. To the best of our knowledge, Li et al. [58] proposed the first CNN-based MFIF method, which utilizes a CNN learning a decision map to generate fused images. Later, serval decision-based MFIF methods are proposed, such as MCNN [21], HF-Seg [59], MMF-Net [60] and SSAN [61]. In end-to-end MFIF methods, the fused image is learned directly through training without post-processing steps. Several encoder–decoder architectures are proposed, such as IFCNN [62] and U2Fusion [63].

### 2.3. Self-Attention-Based Backbone

The transformer architecture [24] based on the self-attention mechanism has achieved SOTA performance for various tasks in the natural language processing (NLP) domain [64]. Building upon the transformer's success in the NLP domain, Dosovitskiy et al. [25] introduced the vision transformer (ViT), which has demonstrated excellent results as an alternative to convolutional networks while requiring significantly fewer computational resources for training. Most recently, several excellent works based on ViTs [30] have emerged, notably Liu et al. [31] proposed a hierarchical transformer known as the Swin Transformer. This architecture offers flexibility in modeling at various scales, exhibits linear computational complexity concerning image size and is compatible with a wide range of vision tasks. Building on the Swin Transformer architecture, Cao et al. [33] prosed the pure transformer-based U-shaped encoder–decoder network (Swin-Unet).

## 3. Methodology

### 3.1. Overall Architecture

We proposed an end-to-end MFIF module for microscope images based on the Swin Transformer architecture, as presented in Figure 1. The features pyramid of the multi-focus images was encoded by the Swin-Transformer-based model Swin-S [31]. To generate the fused image, a Swin-Transformer-based decoder with patch expansion (Figure 1) was applied to decode the feature pyramid. A Swin Transformer block is composed of a LayerNorm (LN) layer, a multi-head self-attention module, a residual connection and a two-layer MLP with GELU non-linearity. The window-based multi-head self-attention (*W-MSA*) module and the shifted window-based multi-head self-attention (*SW-MSA*) module are applied in the two successive transformer blocks (Figure 2). Based on such a window partitioning mechanism, the Swin Transformer blocks are computed as:

$$\hat{z}^l = W - MSA(LN(z^{l-1})) + z^{l-1} \tag{1}$$

$$z^l = MLP(\hat{z}^l) + \hat{z}^l \tag{2}$$

$$\hat{z}^{l+1} = SW - MSA(LN(z^l)) + z^l \tag{3}$$

$$z^{l+1} = MLP(\hat{z}^{l+1}) + \hat{z}^{l+1} \tag{4}$$

where $\hat{z}^l$ and $z^l$ denote the ouputs of the *(S)W-MSA* module and the MLP module of the $l^{th}$ block, respectively. Self-attention is computed as:

$$Attention(Q, K, V) = SoftMax(\frac{QK^T}{\sqrt{d}} + B)V \tag{5}$$

where $Q, K, V \in \mathbb{R}^{M^2 \times d}$ represent the query, key and value matrices. $M^2$ and $d$ denote the number of patches in a window and the dimension of the query or key, respectively. And the values in B are taken from the bias matrix $\hat{B} \in \mathbb{R}^{(2M-1) \times (2M+1)}$.

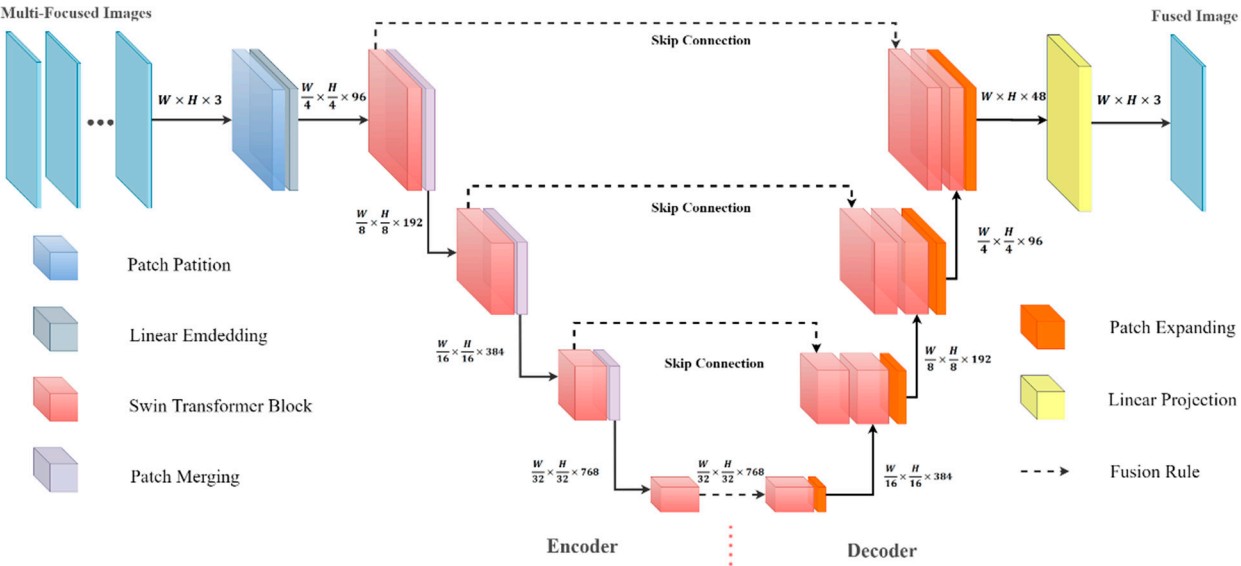

**Figure 1.** The architecture of the end-to-end MFIF models based on the Swin Transformer block, U-Swin fusion model.

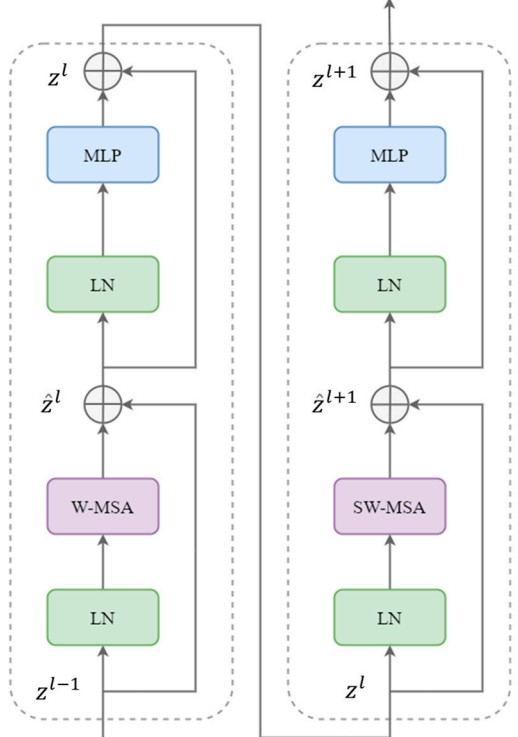

**Figure 2.** Two successive Swin Transformer blocks.

Initially, the input RGB images were divided into non-overlapping patches with a $4 \times 4$ patch and a linear embedding layer and were applied to project raw-value features into an arbitrary dimension, similar to ViTs [25]. To generate hierarchical representations, several Swin Transformer blocks and patch-merging layers were applied to encode the transformed patch tokens. The patch-merging layer concatenates the features of each group of $2 \times 2$ neighboring patches to down-sample and increase dimensions. Swin Transformer blocks are applied to learn feature representations without dimension transformation. To generate the fused image, symmetric Swin Transformer blocks and patch expansion were applied to perform up-sampling in U-Swin fusion models, like Swin-Unet [32]. The patch expansion layer reshapes the feature maps into a higher resolution feature map ($2\times$ up-sampling). To fuse the hierarchical features of multi-input images, varieties of fusion rules (tensor max and tensor mean) have been utilized to fuse the transformed path tokens, as presented in the IFCNN [62]. Furthermore, two types of mix fusion rules (simple mix and channel mix) were used in this study, which can be expressed as follows:

$$F = \omega_1 F^{Max} + \omega_2 F^{Mean} \ (Simple \ mix) \tag{6}$$

$$F_i = \omega_{(1, \, i)} F_i^{Max} + \omega_{(2, \, i)} F_i^{Mean} \ (Channel \ mix) \tag{7}$$

where $F^{Max}$ and $F^{Mean}$ denote the fused features from tensor max and tensor mean, respectively, and i represents the different channel of features. $\omega$ is not the hyperparameter that can be optimized during training mode. In this study, the channel mix fusion rule was implemented by the depthwise convolution to avoid information exchange between different channels. Inspired by U-net [65], the skip connections were applied to concatenate the hierarchical features from the encode with the up-sample features to enhance overall performance. The shallow features are concatenated with the up-sampling features from deep features to reduce the loss of spatial information caused by down-sampling.

### 3.2. Training Data Augmentation

As reported in the previous literature, the multi-focus image dataset can be more easily generated from other types of image datasets, and more importantly, the ground-truth fusion images of the multi-focus images could be obtained simultaneously while generating training data [62]. In this paper, 525 texture images were selected from the Describable Textures Datasets [66], Salzburg Texture Image Database (https://www.wavelab.at/sources/STex/ (accessed on 19 April 2023)) and Original Brodatz's Texture Database (http://multibandtexture.recherche.usherbrooke.ca/original_brodatz.html (accessed on 19 April 2023)). The texture images contain bubbly, fibrous, honeycombed, marble, pleated, veined, etc., images as well as images of objects such as gravel and hair. These textures are similar to the biomedical images observed under a microscope. Initially, the RGB images were resized to $224 \times 224$ by cubic interpolation. Depth images for each texture dataset were generated by the Perlin noise algorithm with specified frequencies (2, 4, 8) and random curves (from 1 to 7). As a result, we generated 1575 training datasets. The resized pairs of RGB images ($I_s$, Figure 3a) and depth images of the datasets ($I_d$, Figure 3b) were applied to create the multi-focus image dataset as the following procedures:

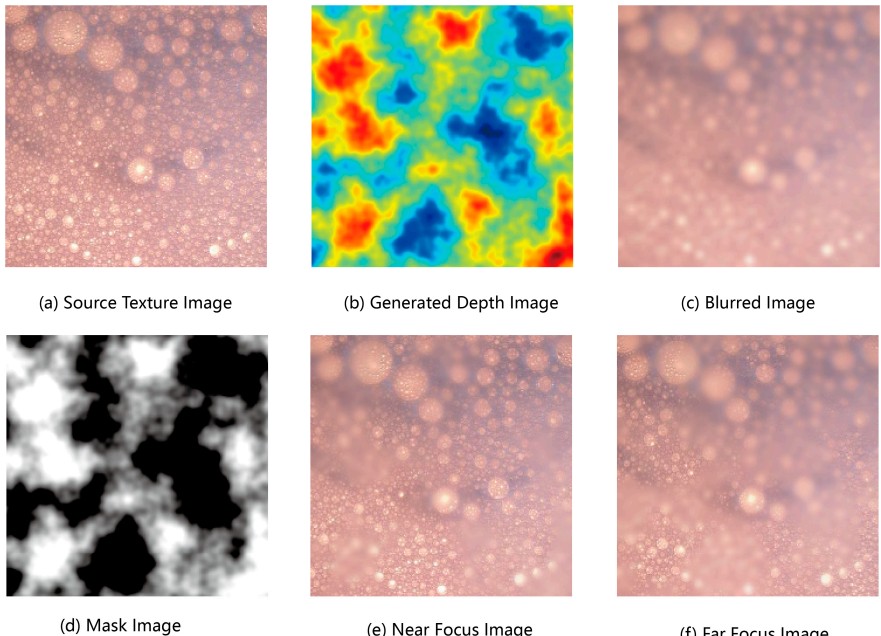

(a) Source Texture Image     (b) Generated Depth Image     (c) Blurred Image

(d) Mask Image     (e) Near Focus Image     (f) Far Focus Image

**Figure 3.** Example of generating multi-focus training dataset according to Section 3.2. (**a**,**b**) are the source image from texture datasets and depth image generated by Perlin noise algorithm, respectively. (**c**) denotes the randomly blurred image. (**d**) shows the mask map according to random threshold depth. (**e**,**f**) are the near-focus image and far-focus image, which are the input images for training models.

1. A completely blurry image ($I_b$, Figure 3c) is generated by randomly blurring the source RGB image ($I_s$) with a Gaussian filter, which can be expressed as:

$$I_b = G * I_s \tag{8}$$

Here, * denotes the convolution operation, and G denotes the Gaussian kernel, which is generated with a random kernel radius, *kr*, from 1 to 15:

$$G(x,y) = \frac{1}{\sqrt{2\pi}\sigma} e^{\frac{x^2+y^2}{2\sigma^2}} \tag{9}$$

where σ denotes the standard deviation of the Gaussian filter and can be expressed as:

$$\sigma = 0.3 \times (kr - 1) + 0.8 \tag{10}$$

2. The mask map ($I_m$, Figure 3d) was generated from depth images of datasets ($I_d$) based on a random threshold depth ($d_{th}$).

$$I_m(x,y) = \begin{cases} 0, & I_d(x,y) \geq d_{th} \\ 1, & I_d(x,y) < d_{th} \end{cases} \tag{11}$$

$$d_{th} = \gamma \times (max(I_d) - min(I_d)) + min(I_d) \tag{12}$$

Here, $\gamma$ denotes the depth threshold ratio, which is randomly selected from the range 0.3 to 0.7.

3.     The near-focus image $I_n$ (Figure 3e) and the far-focus image $I_f$ (Figure 3f) were generated based on source image ($I_s$), completely blurred image ($I_b$) and the mask map ($I_m$), which can be expressed as the following equation:

$$\begin{cases} I_n = I_s \cdot I_m + I_b \cdot (1 - I_m) \\ I_f = I_b \cdot I_m + I_s \cdot (1 - I_m) \end{cases} \tag{13}$$

Naturally, the source images ($I_s$) can serve as the ground truth for fused images.

*3.3. Training Details*

The total of training process can be divided into two parts which are described as the following:

Stage 1: To improve the generation of hierarchical features, the encode part of the fused model was initialized by the pretrained model Swin-S [31], which has been trained on ImageNet-1K dataset [67] for image classification. Subsequently, the fusion model was trained on the training dataset generated above with mean squared loss and the AdamW optimizer [68] for 1000 epochs. It is worth noting that we employed a cosine decay learning rate with an 0.0005 initial learning rate, 0.05 weight decay and 20 epochs of linear warm-up. The batch size during the training period was set to 16.

Stage 2: Mean square error loss is the basic loss function which is used to regularize the prediction close to the ground-truth output. To encourage the network to produce images with greater texture similarity to the ground-truth fusion images, we incorporated a texture loss into the training model with the same hyperparameters as in the previous stage (1000 epochs, 16 batch size and Adam optimizer). The training loss is expressed as:

$$F_{loss} = \omega_1 M_{loss} + \omega_2 T_{loss} \tag{14}$$

Here, $M_{loss}$ and $T_{loss}$ denote the mean square error and textures loss separately. In this study, $\omega_1$ and $\omega_2$ are both set to 1.

**4. Experiments**

In this section, we conducted extensive experiments to validate the advantage of the proposed fusion models based on various evaluation methods. Firstly, the basic experimental settings are described, and then qualitative and quantitative results are illustrated and discussed for the EDoF Fraunhofer dataset [69].

*4.1. Experiment Details*

To evaluate the effectiveness of the transformer architecture in multi-focus image fusion, we compared the fusion performance of our Swin-Transformer fusion model with representative end-to-end methods.

We conducted comparisons with the SOTA unsupervised deep learning fusion model, called FusionDN [70], which employs a unified densely connected network to fuse images. Additionally, our models applied the same fusion rules as the IFCNN [62], which is a representative fusion framework based on the convolutional neural network. Therefore, we compared our fusion models with the IFCNN framework with the elementwise-maximum, elementwise-mean and elementwise-sum fusion rules, referred to as IFCNN-MAX, IFCNN-MEAN and IFCNN-SUM, respectively. Furthermore, we used the fusion result from four existing fusion models for validation: an unsupervised adversarial network with adaptive and gradient joint constrains (MMF-GAN) [71], a fast unified network based on proportional maintenance of gradient and intensity (PMGI) [72], a novel unified and unsupervised end-to-end image fusion network (U2Fusion) [63] and a novel general image fusion framework based on cross-domain long-range learning and the Swin Transformer (Swin fusion) [73]. Moreover, we conducted a comparison with recent SOTA fusion models, a novel memory unit architecture for image fusion (MUFusion) [74] and one of the first zero-shot models for image fusion (ZMFF) [75].

In this study, the complex wavelet extended-depth-of-field method, which remains the traditional gold standard method for image fusion [76], was utilized to generate the ground truth for multi-focus microscope image fusion. To perform image fusion with this method, we used ImagJ (v1.54b) software [77] with the extended-depth-of-field plugin (http://bigwww.epfl.ch/demo/edf/ (accessed on 25 May 2023)).

To evaluate the performance of the different models, we employed three widely used metrics: mean square error (*MSE*), peak signal-to-noise ratio (*PSNR*) and structure similarity index (*SSIM*). *MSE* and *PSNR* are commonly used metrics for objective image quality assessment, which can be expressed as follows:

$$MSE = \frac{1}{mn} \sum_{i=0}^{m-1} \sum_{j=0}^{n-1} [GT(i,j) - F(i,j)]^2 \tag{15}$$

$$PSNR = 10 \cdot log_{10}(\frac{255^2}{MSE}) \tag{16}$$

*MSE* can quantify the discrepancy between the ground truth and fused image at the pixel level, while *PSNR* represents the ratio between effective information and noise in the fused image. However, *MSE* and *PSNR* do not account for the spatial arrangement of the pixels, which is essential for image quality assessment [78]. Moreover, *MSE* and *PSNR* are easily dominated by local outliers. To overcome the weakness of *MSE/PSNR*, a new metric, named the structural similarity (*SSIM*) index, has been developed to consider location knowledge during a quality assessment experiment [79], which can be expressed as:

$$SSIM = L(GT, F) \cdot C(GT, F) \cdot S(GT, F) \tag{17}$$

Here, *L*, *C* and *S* correspond to luminance, contrast and structural similarity, respectively.

### 4.2. Experiment Results for EDoF Fraunhofer Dataset

Many end-to-end fusion models are limited to the fusion of only two images, which poses a challenge when dealing with multi-focus microscope images that consist of a series of z-slices. To ensure a fair evaluation of fusion model performance, we selected two microscope images from the EDoF Fraunhofer dataset. Since ZMFF is a zero-shot image fusion model and the model should be trained for each fusion step, we evaluated the results for different training epochs (600 epochs, 900 epochs and 1300 epochs). The comparative fusion examples generated by various models are presented in Figure 4. Significantly, the fused image generated by Fusion DN exhibits conspicuous errors along its edges, indicating a notable shortfall in edge fidelity. Furthermore, the fused images resulting from PMGI and U2Fusion display a discernible reduction in luminance, while PMGI manifests pronounced and undesirable shadow artifacts. In contrast, the images produced by MFF-GAN, Swin Fusion and MUFusion exhibit higher luminance when compared to both the ground truth generated from the complex wavelet extended-depth-of-field method and the source input images. Surprisingly, the ZMFF model achieved satisfactory performance for multi-focus microscope image fusion after 500 epochs of training without priors learned from large-scale datasets. Notably, IFCNN and U-Swin, each employing distinct fusion rules, excel in generating fusion images that are universally regarded as superior in terms of quality. This subjective assessment underscores their prominence in achieving the highest levels of image fusion quality.

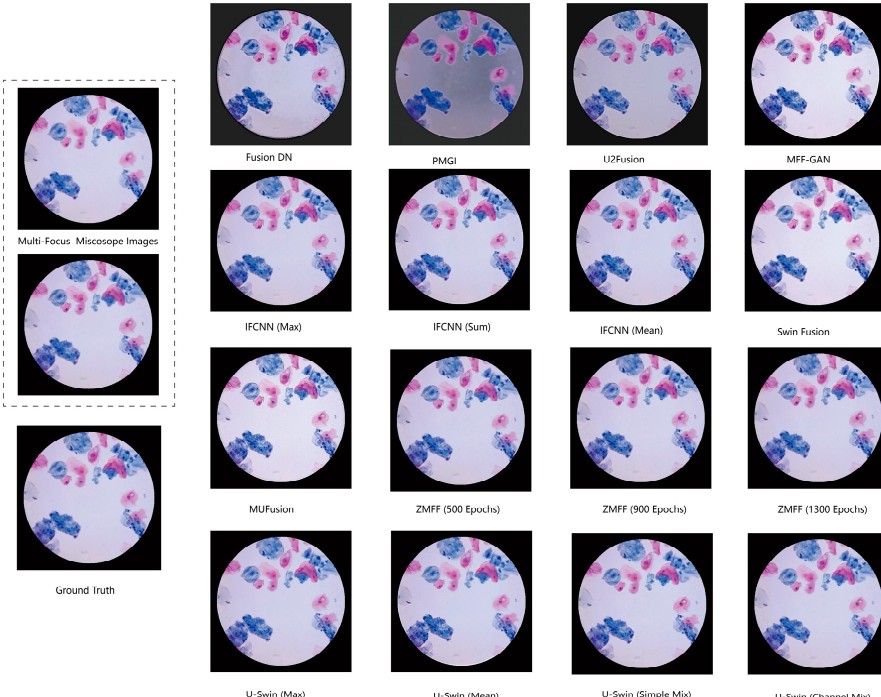

**Figure 4.** The comparison example of two selected multi-focus fused images from EDoF Fraunhofer dataset.

Table 1 presents the quantitative evaluation results for the EDoF Fraunhofer dataset. The values highlighted in bold font represent the best results in their respective evaluation metric columns. These values represent the mean and standard deviation of the evaluation metrics across the entire dataset. The statistics evaluation results listed in Table 1 clearly indicate that our proposed U-Swin fusion model, particularly when employing channel mix fusion, achieves the best performance in terms of mean squared error (*MSE*) and ranks second in structural similarity (*SSIM*). Notably, the fused images from MFF-GAN and Swin fusion exhibit relatively lower peak signal-to-noise ratios (*PSNRs*), which correlates with comparable performance in *SSIM*. In contrast, MUFusion cannot achieve optimal performance for multi-focus microscope image fusion. Surprisingly, the ZMFF model achieved comparable performance to U-Swin (mean) and IFCNN (mean) without prior training from large-scale datasets. However, Since ZMFF needs to be retrained every time image fusion is performed, this will take a lot of time. It is worth noting that the other fusion models, with the exception of IFCNN and ZMFF, struggle to generate satisfactory fusion images. This difficulty is often attributed to the characteristics of microscope images, which typically exhibit low contrast and repetitive structures that pose challenges for feature extraction. The majority of fusion models, with the notable exceptions of IFCNN and the U-Swin fusion model, conduct the fusion process within the Y component of the YCrCb color space. This approach, while effective for many types of images, may not be optimally suited for microscope images. The Y component primarily encodes the brightness information, which is important for overall image quality but may not adequately capture the nuances of microscope images. As a result, these models tend to yield less favorable results when assessed using metrics like mean squared error (*MSE*) and peak signal-to-noise ratio (*PSNR*). The limitations arise from the fact that microscope images often exhibit unique characteristics such as low contrast, repetitive structures and lower signal-to-noise ratios, which demand more sophisticated fusion techniques for accurate and high-quality results.

**Table 1.** Quantitative evaluation results for EDoF Fraunhofer dataset for all images. † means the evaluation results from our proposed model. The values highlighted in bold font represent the best results in their respective evaluation metric columns.

|  | MSE | PSNR | SSIM |
| --- | --- | --- | --- |
| FusionDH | 1129.10 ± 515.22 | 18.07 ± 2.03 | 0.6034 ± 0.0115 |
| PMGI | 5185.17 ± 1817.41 | 11.31 ± 1.80 | 0.5086 ± 0.0359 |
| U2Fusion | 586.70 ± 204.62 | 20.73 ± 1.63 | 0.6543 ± 0.0053 |
| MFF-GAN | 137.00 ± 24.77 | 26.85 ± 0.90 | 0.9690 ± 0.0085 |
| IFCNN (Max) | 4.99 ± 0.81 | 41.21 ± 0.67 | **0.9844 ± 0.0022** |
| IFCNN (Sum) | 9.03 ± 1.41 | 38.62 ± 0.66 | 0.9779 ± 0.0027 |
| IFCNN (Mean) | 12.31 ± 1.93 | 37.28 ± 0.66 | 0.9768 ± 0.0030 |
| Swin Fusion | 173.41 ± 48.72 | 25.96 ± 1.48 | 0.9785 ± 0.0032 |
| MUFusion | 150.60 ± 44.95 | 26.53 ± 1.22 | 0.8366 ± 0.0068 |
| ZMFF (600 Epochs) | 24.89 ± 7.93 | 34.32 ± 1.08 | 0.9701 ± 0.0049 |
| ZMFF (900 Epochs) | 17.18 ± 5.43 | 35.97 ± 1.33 | 0.9749 ± 0.0037 |
| ZMFF (1300 Epochs) | 14.59 ± 3.54 | 36.66 ± 1.34 | 0.9761 ± 0.0033 |
| U-Swin (Max) † | 5.36 ± 2.05 | 41.17 ± 1.76 | 0.9829 ± 0.0030 |
| U-Swin (Mean) † | 15.78 ± 3.60 | 36.26 ± 0.96 | 0.9651 ± 0.0070 |
| U-Swin (Simple Mix) † | 7.88 ± 2.67 | 39.44 ± 1.61 | 0.9745 ± 0.0046 |
| U-Swin (Channel Mix) † | **2.74 ± 0.63** | **43.85 ± 0.93** | 0.9839 ± 0.0020 |

Figure 5 provides a visual illustration of the comparison between multi-fused images for all slices within the EDoF Fraunhofer dataset. In cases where the fusion models are designed to accommodate only double-image fusion, generating a fully focused image involved a step-by-step process. It is worth noting that ZMFF cannot obtain effective results as the number of fused images increases. This may be due to the fact that for a series of multi-focus microscopic images, the multi-focus images have a high degree of similarity, which increases the difficulty of zero-shot model optimization. The results produced by the FusionDN, PMGI and U2Fusion methods exhibit a noticeable and substantial misalignment when compared against the ground truth. With an increase in the number of fused images, the cumulative fused error becomes more pronounced for models designed to support only the fusion of two images. As a result of these pronounced disparities, there appears to be no necessity for a quantitative evaluation of these models. The imagery generated by MFF-GAN and MUFusion prominently showcases synthetic textures that are readily discernible to the observer. Conversely, our proposed U-Swin fusion model has demonstrated exceptional performance, particularly for models utilizing channel mix fusion rules, as emphasized in Table 2.

**Table 2.** Quantitative evaluation results for EDoF Fraunhofer dataset for two-image fusion. † means the evaluation results from our proposed model. The values highlighted in bold font represent the best results in their respective evaluation metric columns.

|  | MSE | PSNR | SSIM |
| --- | --- | --- | --- |
| MFF-GAN | 217.62 ± 60.80 | 24.95 ± 1.36 | 0.7340 ± 0.0284 |
| IFCNN (Max) | 7.15 ± 1.38 | 39.66 ± 0.80 | **0.9772 ± 0.0030** |
| IFCNN (Mean) | 12.711 ± 1.93 | 37.28 ± 0.66 | 0.9768 ± 0.0030 |
| Swin Fusion | 208.57 ± 50.52 | 25.10 ± 1.28 | 0.9660 ± 0.0051 |
| MUFusion | 345.51 ± 159.47 | 23.17 ± 1.91 | 0.7405 ± 0.0293 |
| U-Swin (Max) † | 8.60 ± 3.09 | 39.09 ± 1.70 | 0.9762 ± 0.0040 |
| U-Swin (Mean) † | 54.10 ± 8.09 | 30.84 ± 0.64 | 0.8911 ± 0.0132 |
| U-Swin (Simple Mix) † | 8.91 ± 2.73 | 38.84 ± 1.41 | 0.9708 ± 0.0053 |
| U-Swin (Channel Mix) † | **6.28 ± 1.59** | **40.29 ± 1.07** | 0.9649 ± 0.0036 |

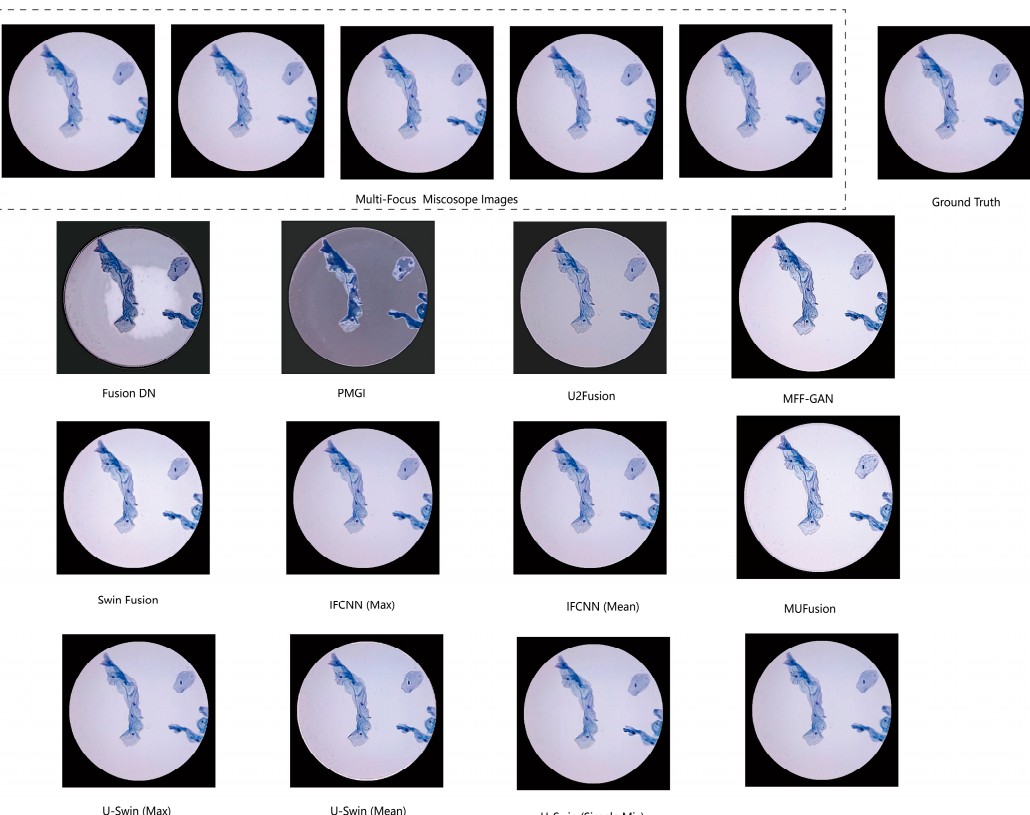

**Figure 5.** The comparison example of multi-focus fused images from EDoF Fraunhofer dataset.

*4.3. Experiment Results for Various Color Spaces*

In order to assess the effectiveness of our proposed model based on the Swin Transformer backbone, we explored various color spaces (RGB, YCrCb, HSV, LAB) for multifocus microscope image fusion. We evaluated the performance of the fusion model for the two selected microscope images from the EDoF Fraunhofer dataset, as discussed in Section 4.2. In this section, we conducted qualitative and quantitative evaluations of the U-Swin fusion model with the channel mix fusion rule. The proposed model shown in previous sections fused the microscope images in RGB color spaces, treating the three channels of the multi-focus microscope images as the model inputs. Figure 6 and Table 3 demonstrate that our proposed model with the channel mix fusion rule could achieve satisfactory performance in various color spaces. The outcome suggests that the proposed model (U-Swin fusion model) achieves state-of-the-art (SOTA) results for the RGB color space and is applicable to other color spaces as well.

**Table 3.** Quantitative evaluation results for the U-Swin fusion model with channel mix fusion rule for various color spaces. The values highlighted in bold font represent the best results in their respective evaluation metric columns.

|  | MSE | PSNR | SSIM |
|---|---|---|---|
| RGB | **2.74 ± 0.63** | **43.85 ± 0.93** | **0.9839 ± 0.0020** |
| YCrCb | 4.39 ± 0.66 | 41.75 ± 0.63 | 0.9526 ± 0.0025 |
| HSV | 6.40 ± 1.27 | 40.16 ± 0.86 | 0.9623 ± 0.0052 |
| LAB | 4.96 ± 0.78 | 41.23 ± 0.67 | 0.9628 ± 0.0021 |

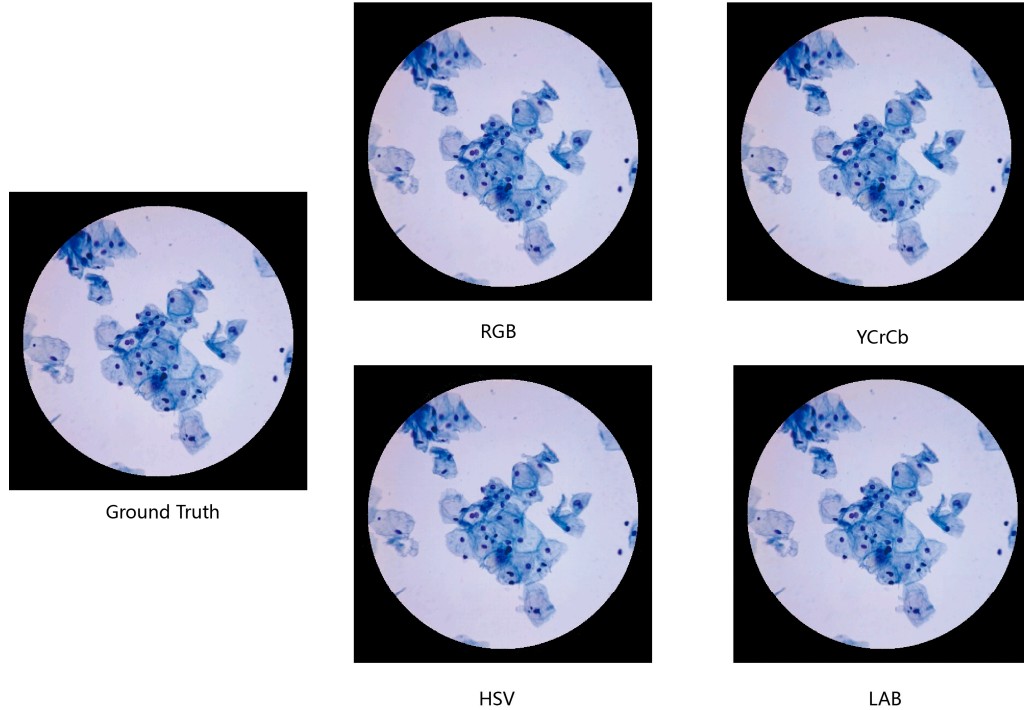

**Figure 6.** The comparison example of the U-Swin fusion model with channel mix fusion rule for various color spaces.

## 5. Conclusions

In this paper, we present a groundbreaking end-to-end microscope image fusion model, built on the Swin Transformer backbone, known as the U-Swin fusion model. Leveraging the innate capabilities of transformers to capture long-range dependency features, our innovative transformer-based models consistently produce fusion images that either match or surpass the state-of-the-art (SOTA) image fusion algorithms. Additionally, they exhibit remarkable generalization ability for fusing multi-focus microscope images. Notably, the pure transformer-based U-Swin fusion model, incorporating channel mix fusion rules, attains superior performance in numerous evaluation metrics compared to the majority of existing end-to-end fusion models. This work lays a pioneering foundation for applying transformer-based networks within the realm of microscope image fusion. Despite the extensive experimental results that validate the advantages of our proposed models, there remain several avenues for improvement in pursuit of even more robust image fusion models. For instance, exploring complex transformer backbones and fusion rules holds promise for further enhancing the model's performance in the future.

**Author Contributions:** Conceptualization, H.G. and H.S.; Methodology, H.H.X.; Formal analysis, H.H.X. and K.G.; Investigation, W.L.; Data curation, H.H.X.; Writing—original draft, H.H.X.; Writing—review & editing, H.G.; Funding acquisition, H.S. All authors have read and agreed to the published version of the manuscript.

**Funding:** This research was funded by Key R&D Program of Zhejiang Province (2021C01016) and National Natural Science Foundation of China (61827805). And the APC was funded by Jiaxing Key Laboratory of Visual Big Data and Artificial Intelligence, Zhejiang Province, China.

**Institutional Review Board Statement:** Not applicable.

**Informed Consent Statement:** Not applicable.

**Data Availability Statement:** The data presented in this study are available in article.

**Conflicts of Interest:** The authors declare no conflict of interest.

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
