# Peer review of "Multi-Focus Microscopy Image Fusion Based on Swin Transformer Architecture"

_applsci, doi:10.3390/app132312798_

Round 1

Reviewer 1 Report

Comments and Suggestions for Authors

Authors introduce the U-Swin Fusion Model that is an effective and efficient Transformer-based architecture designed for the fusion of multi-focus microscope images. They claimed that their modulators demonstrated superior capability for multi-focus images fusion achieve comparable or even better fusion images than the exists state-of-the-art image fusion algorithms and demonstrates adequate generalization ability for multi-focus microscope images fusion. The work is good but existing method is used.

Q1. Why Swin-Transformer with shifted window is used?

Q2. Specify the function and advantages of multi-focus image fusion.

Q3. Swin-Transformer backbone- It is an existing model and it is used in image fusion. The long range needs specified exact value.

Q4. Specify the new concept used in the paper.

Q5. There are no comparisons with existing methods.

Comments on the Quality of English Language

-

Reviewer 2 Report

Comments and Suggestions for Authors

the fusion models designed to accommodate only double image fusion, such as FusionDN, PMGI, and U2Fusion, exhibit noticeable and substantial misalignment when compared to the ground truth. This misalignment becomes more pronounced with an increase in the number of fused images. As a result, there appears to be no necessity for a quantitative evaluation of these models.

 The u-Swin Fusion Model conducts the fusion process within the Y component of the YCrCb color space, which may not be optimally suited for microscope images. Microscope images often exhibit unique characteristics such as low contrast, repetitive structures, and lower signal-to-noise ratios, which demand more sophisticated fusion techniques for accurate and high-quality results. The limitations arise from the fact that the Y component primarily encodes brightness information, which is important for overall image quality but may not adequately capture the nuances of microscope images.

Additionally, while the article presents groundbreaking end-to-end microscope image fusion models, there are still avenues for improvement. Exploring complex transformer backbones and fusion rules holds promise for further enhancing the model's performance in the future.

Provide a more comprehensive evaluation of the fusion models: While the article mentions that there is no necessity for a quantitative evaluation of certain fusion models, it would be beneficial to include a qualitative analysis of their performance. This could involve comparing the fused images against the ground truth using visual inspection and subjective assessment.

Explore alternative color spaces for microscope image fusion: Since microscope images have unique characteristics, such as low contrast and repetitive structures, it would be valuable to investigate other color spaces that may better capture these nuances. For example, the Lab color space separates the luminance and chrominance components, which could potentially improve the fusion results for microscope images.

Investigate more advanced fusion techniques: While the article presents state-of-the-art end-to-end fusion models, there is room for exploring more sophisticated techniques. This could involve incorporating complex transformer backbones or fusion rules to further enhance the performance and accuracy of the fusion models.

Consider the impact of noise and artifacts: Microscope images often suffer from noise and artifacts, which can affect the quality of the fused images. It would be beneficial to address these challenges and propose methods to mitigate their impact on the fusion process. This could involve denoising techniques or artifact removal algorithms.

Include a comparison with existing fusion methods: To provide a more comprehensive analysis, it would be valuable to compare the proposed fusion models against existing fusion methods. This could involve benchmarking the performance of the models against well-established fusion algorithms and highlighting their advantages and limitations.

Reviewer 3 Report

Comments and Suggestions for Authors

The authors introduce the U-Swin Fusion Model, a Transformer-based architecture designed for the fusion of multi-focus microscope images. The Swin-Transformer with shifted window and path merging is used as the encoder to extract hierarchical context features, while a Swin-Transformer-based decoder with patch expanding is designed to perform the un-sampling operation, generating the fully focus image. Skip connections are applied to concatenate the hierarchical features from the encoder with the decoder up-sample features to enhance the performance of feature decoder. The authors created a substantial dataset of multi-focus images, primarily derived from texture datasets, to facilitate comprehensive model training. The authors demonstrate that the U-Swin Fusion Model delivers optimal performance among most existing end-to-end fusion models for multi-focus microscope image fusion.

Major Comments

The authors should provide more details on the dataset used for training and testing the U-Swin Fusion Model. For example, what is the size of the dataset, how many classes are there, and how was the dataset collected?

The authors should provide more details on the experimental setup, such as the hyperparameters used for training the U-Swin Fusion Model, the number of epochs, and the batch size.

The authors should compare the performance of the U-Swin Fusion Model with more state-of-the-art methods for multi-focus microscope image fusion, not just "most existing end-to-end fusion models."

Modifications of introduction “Discovering novel soliton solutions for (3+ 1)-modified fractional Zakharov–Kuznetsov equation in electrical engineering through an analytical approach” “Fractional-order modeling: Analysis of foam drainage and Fisher's equations” “Probing families of optical soliton solutions in fractional perturbed Radhakrishnan–Kundu–Lakshmanan model with improved versions of extended direct algebraic method”

Minor Comments

The authors should provide more details on the channel mix fusion rules used in the U-Swin Fusion Model.

The authors should provide more details on the Swin-Transformer with shifted window and path merging used as the encoder to extract hierarchical context features.

The authors should provide more details on the Swin-Transformer-based decoder with patch expanding used to perform the un-sampling operation.

Questions

What is the Swin-Transformer with shifted window and path merging used for in the U-Swin Fusion Model?

How are skip connections applied in the U-Swin Fusion Model?

What is the dataset used for training and testing the U-Swin Fusion Model?

How does the U-Swin Fusion Model compare to more state-of-the-art methods for multi-focus microscope image fusion?

What are the channel mix fusion rules used in the U-Swin Fusion Model?

Round 2

Reviewer 2 Report

Comments and Suggestions for Authors

I have given 8 comments on the article while I have received responses against 5 comments. 

Second, out of 5 replies, authors have considered my suggestions as future work. I have checked and considered the replies of the authors but found them not satisfactory. 

Authors are requested to give their reply against each comment plus incorporate comments and if not applicable at this stage then please give solid rational. 

Reviewer 3 Report

Comments and Suggestions for Authors

Accept

Author Response

Thanks for your assistance in revising and publishing the article.

Round 3

Reviewer 2 Report

Comments and Suggestions for Authors

Dear Authors

I have again checked your comments and read them thoroughly. Now your comments are satisfactory.

The article is now ready to publish.